# What Is Driving the Growth of Inorganic Glass in Smart Materials and Opto-Electronic Devices?

**DOI:** 10.3390/ma14112926

**Published:** 2021-05-29

**Authors:** Daniel Alves Barcelos, Diana C. Leitao, Laura C. J. Pereira, Maria Clara Gonçalves

**Affiliations:** 1Departamento de Engenharia Química, Instituto Superior Técnico, Universidade de Lisboa, Av. Rovisco Pais, 1049-001 Lisboa, Portugal; danielbarcelos@tecnico.ulisboa.pt; 2CQE, Centro de Química Estrutural, Av. Rovisco Pais, 1049-001 Lisboa, Portugal; 3INESC Microsistemas e Nanotecnologias, R. Alves Redol 9, 1000-029 Lisboa, Portugal; dleitao@inesc-mn.pt; 4Departamento de Física, Instituto Superior Técnico, Universidade de Lisboa, Av. Rovisco Pais, 1049-001 Lisboa, Portugal; 5Departamento de Engenharia e Ciências Nucleares, Instituto Superior Técnico, Universidade de Lisboa, 2685-066 Bobadela LRS, Portugal; lpereira@ctn.tecnico.ulisboa.pt; 6Centro de Ciências e Tecnologias Nucleares, Instituto Superior Técnico, Universidade de Lisboa, 2685-066 Bobadela LRS, Portugal

**Keywords:** inorganic glass, smart materials, opto-electronic devices

## Abstract

Inorganic glass is a transparent functional material and one of the few materials that keeps leading innovation. In the last decades, inorganic glass was integrated into opto-electronic devices such as optical fibers, semiconductors, solar cells, transparent photovoltaic devices, or photonic crystals and in smart materials applications such as environmental, pharmaceutical, and medical sensors, reinforcing its influence as an essential material and providing potential growth opportunities for the market. Moreover, inorganic glass is the only material that is 100% recyclable and can incorporate other industrial offscourings and/or residues to be used as raw materials. Over time, inorganic glass experienced an extensive range of fabrication techniques, from traditional melting-quenching (with an immense diversity of protocols) to chemical vapor deposition (CVD), physical vapor deposition (PVD), and wet chemistry routes as sol-gel and solvothermal processes. Additive manufacturing (AM) was recently added to the list. Bulks (3D), thin/thick films (2D), flexible glass (2D), powders (2D), fibers (1D), and nanoparticles (NPs) (0D) are examples of possible inorganic glass architectures able to integrate smart materials and opto-electronic devices, leading to added-value products in a wide range of markets. In this review, selected examples of inorganic glasses in areas such as: (i) magnetic glass materials, (ii) solar cells and transparent photovoltaic devices, (iii) photonic crystal, and (iv) smart materials are presented and discussed.

## 1. Introduction

The growth of the world population followed by economic prosperity and rise in disposable income in emerging economies [1,2] led to a significant increase in the consumption of smart materials and opto-electronic devices, boosting a global inorganic glass demand [3,4,5,6,7]. Further, the dynamically changing technology landscape encourages consumers to shift to electronics integrated with the latest technologies. This leads to a growth in the consumer electronics market and demands a rapid reduction in the product lifecycle.

Inorganic glass remains a pivotal material in many scientific and engineering applications, including optics, photonics, opto-electronics, photovoltaics, hermetic seals, and is also integrated into microfluidic, microelectromechanical (MEMs), and chemical, environmental, pharmaceutical, medical, or biological sensors [8,9]. Unmatched optical transparency paired with outstanding thermal and chemical resistance makes inorganic glass the first-choice material in many applications in science, industry, and society and a hard product to beat.

Traditionally used as a package, hollow inorganic glass soon spanned into architecture (with flat geometry) as an element that provided cohesion between the inside and the outside, and into terrestrial, maritime, and aerospace transport industries offering protection and security, resistance to thermal shock and to impact from projectiles (as in flat or curved laminated or tempered security glasses) [10]. Latterly used as thin film or coatings, inorganic glass brings a wide diversity of passive functionalities, namely, chemical resistance, customizable hydrophilic or hydrophobic surface character, a wide range of wavelengths control, as well as active functionalities such as bactericidal, self-cleaning, light-emission, up-conversion, electrical-switching, photonic-crystal, opto-electronic, or photovoltaic properties, being essential in the smart materials industry [10]. Smart materials are the ones that respond to an external stimuli such as pH, temperature, Eh, electrical, light, or mechanical forces [11,12]. The inorganic glass’s ability to incorporate other industrial offscourings and/or residues as raw materials along with the 100% recyclability place it in the top position when sustainability is the crucial parameter in design.

### Inorganic Glass Technology: Historical Highlights

Egyptian core-forming and successive layers protocols, Mesopotamian millefiori, and Syrian metallic glass-blow techniques were the first handcrafted melting-quenching techniques. At the beginning of the 19th century, industrialization reached glass production in hollow and flat geometries, enabling large-scale melting-quenching automatic processes. Soon after, in the middle of the 20th century, the Pilkington float process was developed. Recently, Corning^®^ created innovative melting-quenching protocols to produce high mechanical resistance glass (Corning^®^Gorilla^®^ Glass) and transparent ultra-thin flexible glass (Corning^®^Willow^®^ Glass), which stand out as the most reliable and most robust glasses produced thus far for optoelectronic devices and smart materials (Figure 1). Flexible high mechanical resistance glass is on the way [13]. Gorilla^®^ Glass is a chemical strengthened glass produced through (thermally activated) ion exchange process. When the (aluminosilicate) glass is dipped in a hot bath of molten potassium salt, it heats up and expands. The high temperature from the bath promotes the lixiviation of the Na^+^ ions out of the glass sheet and allows the ion exchange with K^+^ ions from the bath. Because K^+^ ions are larger than Na^+^, they get packed into the space more tightly. As the glass cools, they get squeezed together in this now-cramped space, and a layer of compressive stress on the surface of the glass is formed. Compared with thermally strengthened glass, the stuffing or crowding effect in chemically strengthened glass results in higher surface compression, making it up to four times stronger (Figure 1a) [13]. Willow^®^ Glass is lightweight, ultra-thin, conformable glass at thicknesses of 100 and 200 μm, 1.3 m wide, and up to 300 m in length. Willow^®^ Glass manufacture relies on Corning^®^’s fusion down-draw process (100% mechanized). This melting-quenching process starts with high-quality raw materials, followed by the production of high purity molten glass, which is continually down-drawn. Eventually, it may overflow through an isopipe to form a sheet with a high-quality surface area. No contact with other solid surfaces in quality area occurs during production (Figure 1b–d) [13].

In these first-generation fabrication methodologies (melting-quenching processes), quenching follows raw materials fusion (Table 1) [10,17,18,19,20]. The melting temperature depends on the system phase diagram, being ~1500 °C for aluminosilicate glasses, ~1200–1300 °C for borate and phosphate, and ~900 °C for fluoride systems. At the melting temperature, the glass mixture needs to stay for at least an hour to ensure complete fusion, glass molten fining, and homogenization.

CVD [21,22,23] and PVD [21,24,25] processes appeared as a second-generation in inorganic glass fabrication. However, until the 1970s, melting-quenching was the dominant glass making procedure. Yet, inorganic glass was manufactured in academia and industry, such as with mimetite nature colloidal chemistry processes [26,27,28]. Sol-gel and solvothermal methodologies are two of those examples from the third-generation fabrication methodologies [29,30,31,32,33,34,35,36,37,38,39,40,41,42,43,44,45,46,47,48,49,50,51,52,53,54,55,56,57,58,59,60].

Particularly, sol-gel is used to synthesize glasses and amorphous materials at relatively low processing temperatures (Table 2). This process is developed by preparing a colloidal suspension (sol), followed by gelation of the sol, and finally removing the liquid existing in fine interconnected channels within the gel (colloidal route). Alternatively, a polymeric route, the most common sol-gel process, undergoes hydrolysis of metallic salts, metal alkoxides, or another complex organometallic and polycondensation to form the gel. The gel is then dried at a low temperature, for instance, ~100 °C. Because the dry gel is still porous, it must be sintered at a temperature around the glass transition (Tg) when a thick glass is produced [26,27,28]. Sol-gel is particularly suitable for the deposition of coatings, but it can also be used to fabricate glasses in bulk, NPs, powder, or fiber forms, which have a structure identical to those melt-quenched glasses with the same compositions. The advantages of the sol-gel methodologies over melting-quenching techniques are vast since a highly pure and homogeneous (at a molecular level) product, processed at low temperatures, is easily obtained. The ability to shape the final product into different topologies, such as monolithic blocks (3D), thin/thick films (2D), flexible glass (2D), powders (2D), fibers (1D), or NPs (0D), allows the covering of different challenging areas from electronics, bio- and medical technology, energy, and environment.

Recently, AM, commonly known as 3D printing, emerged as a novel technology able to deliver structures created in a computer-assisted design program with little human intervention. Regarding glass technology, two main AM approaches are under development—a high temperature method, which is based on traditional melting-quenching techniques, and a low-temperature route that is supported basically on sol-gel methodologies, both integrating the fourth-generation fabrication technologies (Table 3) [61,62,63,64,65,66,67,68,69,70,71,72,73,74].

Inorganic glass is extremely versatile, as it can be molded, formed, blown, plated, sintered, or deposited, thus hitting an ever-growing range of applications in all technological fields. Such processes can take place in a wide range of temperature (down to 100 °C) or can even include AM strategies (Figure 2). The chosen production technique and its particularities dictate the final glass properties and composition.

Today, inorganic glass may be the key material in a growing benefit market addressed to smart materials and optoelectronic devices. From academic start-ups to mature glass manufacturers, a large industrial community is developing novel, unexpected inorganic glass products, taking advantage of the large spectra of inorganic glass compositions and technological methodologies (Table 4 and Table 5). High-tech glass manufacturers, regardless of the employed technology, face challenges of reducing their product development time and time to market to sustain the highly competitive demand. The present work illustrates some of the most recent and unexpected inorganic glass applications on smart materials and optoelectronic devices.

## 2. Glass in Smart Materials and Opto-Electronic Devices

### 2.1. Magnetic Glass Materials

A magnet can be classified as a permanent (hard) or soft material. Permanent magnets have a large coercive field, H_c_, (>1000 A/m) and high magnetic remanence, M_r_, that allows them to retain a large magnetization after being magnetized and consequently to attract and repel other magnets or to magnetize other materials. They can find wide industrial applications such as permanent magnets for motors, manufacturing ultrahigh-density magnetic recording media, and magnetic separation. However, for the inductive applications found, for instance, in power electronics and electrical machines, permanent magnets are not useful. Instead, these applications require soft magnetic materials with lower H_c_ (<1000 A/m) and M_r_, thus they can be easily magnetized and demagnetized. They are suitable for read-write components of magnetic memory devices and cores of transformers and magnetic amplifiers [76].

The global soft magnetic materials market reached a volume of one million USD in 2019, and it will attend a compound annual growth rate (CAGR) of over 10% between 2020 and 2026 (Figure 3).

Inorganic magnetic glasses are a particular class of soft magnetic materials. They usually consist of a glassy matrix in which a magnetic phase crystallizes. This crystalline phase is obtained by controlled crystallization, induced by nucleating agents, of a parent glass containing magnetic ions. The magnetic characterization is usually performed by static (DC magnetization) and dynamic (AC susceptibility) techniques. The theory behind their magnetic properties is briefly referenced with the definition of the key magnetic parameters.

#### 2.1.1. Magnetic Properties

The magnetic moment is usually associated with the orbital motion and/or the spin motion of the charged particles in the material and is a measure of the magnetic field generated by the sample itself.

A magnetic field is a force field such as gravitational and electrical fields characterized by a potential source and expressed by lines that form a contoured sphere. The density of these lines is called magnetic flux density, *B*. In a vacuum, the magnetic field *H* is related with *B* by the permeability of free space, *μ*_0_. When a material is placed in a magnetic field, it can alter the lines of force and therefore modify the flux density, as expressed in Equation (1):(1)$$B=\mu_{0}(H+M)$$

The magnetic behavior of a given material reflects its response to an external magnetic field in its magnetization, *M*. *M* is the sum of all the magnetic moments of a given material per unit of volume or mass. Magnetization depends on size of these magnetic moments and the degree to which they are aligned with respect to each other. The magnetic property that provides a quantitative measure of such response is the magnetic susceptibility, *χ*, defined as:(2)$$\chi =\frac{M}{H}$$

In general, the magnetic susceptibility depends on the electrons that constitute a given system. There are essentially two main contributions for the magnetic susceptibility. The first is diamagnetism, a property inherent to all materials. Diamagnetism reduces the lines of force within the material, which is equivalent to say that it produces a flux opposite to the applied magnetic field that is causing it. The diamagnetic susceptibility is therefore negative, and it does not depend on the field strength or the temperature. Its contribution usually is several orders of magnitude lower than other magnetic contributions.

Second, when there are unpaired electrons in the system, the resulting magnetic moment arises from the spinning and the orbiting of those electrons. If these spins do not interact and are randomly oriented, the application of an external magnetic field tends to align them along the field’s direction. In this sense, a paramagnet concentrates the lines of force, thus increasing the magnetic flux. Since both diamagnetic and paramagnetic materials only exhibit magnetization in the presence of an external field, they are considered as nonmagnetic.

The strength of paramagnetic interactions is temperature dependent. Generally, the bulk magnetic behavior of a material can be described considering how adjacent magnetic moments would interact with each other near zero temperature since, at high temperatures (depending on the composition), all the materials behave as paramagnets due to the effective paramagnetic behavior above their critical temperatures.

When the magnitude of such interactions is significantly higher and propagates in the whole material, it results in a spontaneous magnetization, which is due to the strong exchange interaction between the electron spins throughout the solid. In such cases, a long-range magnetic ordering can occur below a specific temperature and the magnetic transition classified as ferromagnetic (FM) if neighboring spins align parallel, with a characteristic Curie temperature, T_C_, or antiferromagnetic if neighboring spins align antiparallel with a characteristic Néel temperature, T_N_. There is also the ferrimagnetic (FIM) behavior that occurs when the spins are aligned, as in an antiferromagnetic ordering, but with different magnitudes, resulting in a net magnetic moment. A scheme representing these different behaviors is shown in Figure 4.

Ferromagnetic or ferrimagnetic materials exhibit a permanent magnetic moment in the absence of an external field. This spontaneous magnetization is not apparent in materials that are not exposed to an external field due to the presence of small regions called magnetic domains in which the magnetization is in a uniform direction, namely with their spins aligned in the same direction. The magnetization of different domains may point in different directions. Under an external magnetic field, these domains can gradually grow at the expense of the next near domains in a partially reversible process. The boundaries between adjacent magnetic domains are called domain walls. The resulting domain structure is responsible for the characteristic magnetic behavior of such materials. Since the spontaneous magnetization may be several orders of magnitude greater than the applied field, ferromagnetic materials have very high permeabilities, up to 10^6^. When the applied field is removed, a part of the induced domain alignment may be preserved so that the body acts as a permanent magnet. In the case of ferrimagnets, there is an antiferromagnetic coupling between cations occupying different crystallographic sites, and the magnetization of one sublattice is antiparallel to that of another sublattice (Figure 4d). The two magnetizations are of unequal strength, resulting in a net spontaneous magnetization.

In a bulk material, the magnetic field dependence of the magnetization can show hysteresis, as shown in Figure 5b. Hysteresis is observed for ferromagnetic or ferrimagnetic materials below their critical temperature and usually arises from the rearrangement of domain walls within the material.

Consider a sample with randomly aligned domains. By applying an external magnetic field H, the magnetic moments tend to orient in its direction, and as the field increases, the magnetization also increases until it reaches a maximum value, namely magnetic saturation, M_s_, that corresponds to the alignment of all the spins in the sample. As the magnetic field decreases, the magnetization also decreases, but the system retains a degree of magnetization, and the observation of hysteresis appears. The magnetization that remains after all external field is removed is called magnetic remanence, M_r_. The strength of the opposing field required to remove a sample’s magnetization after saturation is the coercive field, H_c_. Repeating the cycle in the opposite direction, from negative to positive magnetic field values, leads to the complete hysteresis loop, as shown in Figure 5a.

The shape of the hysteresis loop depends on how freely the domains walls can rearrange and consequently on physical and chemical properties of the materials (Figure 5b). Materials with large coercive field firmly retain the saturation field when the driving field is removed. These materials, called hard magnets, also exhibit high M_r_ showing a memory effect. On the other hand, soft materials show narrower hysteresis loops.

The thermal behavior of the magnetization of a ferromagnet can also depend on the magnetic field history. If the magnetization does not saturate at a constant value with the applied magnetic field, to determine if the process is irreversible, a set of measurements with the temperature dependence of the magnetization can be performed using zero-field cooling (ZFC) and field cooling (FC). It consists of cooling the sample with H = 0, then applying the desired magnetic field and measuring the magnetization as a function of temperature on heating (ZFC) followed by the same procedure but with an applied magnetic field (FC). The irreversibility of both curves indicates that, below this temperature value, the material behaves as a ferromagnet.

At the nanoscale, ferromagnetic or ferrimagnetic materials with particle size smaller than a specific critical diameter (typically between 3–50 nm) can exhibit different magnetic properties than materials with larger particles since they can randomly change their directions of the magnetization with temperature/time fluctuations below their Curie point. The average magnetization value of such systems in absence of an external magnetic field is close to zero, but in presence of an external magnetic field, nanoparticles align following the magnetic field direction as for paramagnetic materials. However, due to their ferro/ferri-magnetic origin, such nanomaterials show very high magnetic susceptibility, larger than common paramagnetic materials, i.e., they exhibit “super”paramagnetism (SPM) and absence of hysteresis (coercive fields close to zero and very low remanent magnetization, Figure 5c.

In the temperature dependence of the magnetization curves, the presence of a blocking temperature, T_B_, appears at the point of irreversibility of both ZFC and FC curves [79]. Below this value, the magnetic moments are blocked, and the material shows FM or FIM behavior. Above T_B_, the particles behave as superparamagnets. After removing the external field, the magnetic moments of each NPs remain randomly oriented so that the total moment is zero. These thermal and size dependent responses of free NPs are due to the competition of the magnetic anisotropy energy with the thermal energy, thus there is a minimum volume at which the particle remains blocked. Equation (3) represents the Néel-Arrhenius model, where *τ**_N_*, the Néel relaxation time, is the average time between two flips, *τ**_N_* can take values of nano-seconds or even years, *τ*_0_ is the time period (takes values of 10^−9^ or 10^−10^ s), *K* represents the magnetic anisotropy of the particle, *V* the volume of the particle, *T* the temperature, and *k_B_* the Boltzmann constant.
(3)$$\tau_{N}=\tau_{0}e^{\left(\frac{KV}{k_{B}T}\right)}$$

When the magnetization measurement time is less than *τ**_N_*, the magnetization assumes the magnetic moment values of the particles, and the blocked state occurs. On the other hand, when measurement time is much greater than *τ**_N,_* the average value of the measured magnetization appears to be zero, meaning that they are in the superparamagnetic (SPM) state [80]. Typical examples of superparamagnetic NPs are the two common phases of iron oxides, magnetite (Fe_3_O_4_) and maghemite (γ-Fe_2_O_3_).

In real systems of interacting particles, the calculation of the superparamagnetic relaxation time is an extremely complex problem, even in the limit of weak interactions. However, the magnetic dynamics of SPM systems can properly be studied by AC susceptibility measurements. The measurement time is typically 1–100 s for DC measurements and is the inverse of the measurement frequency for AC measurements. The utility of AC susceptibility for superparamagnetism stems from the ability to probe different values of *τ* by varying the measurement frequency. AC susceptibility, *χ*, yields two quantities: the magnitude of the susceptibility, *χ*, and the phase shift, ϕ (relative to the drive signal). Alternately, one can consider the AC susceptibility as having an in-phase, or real, component, *χ*′ = *χ* cos ϕ, and an out-of-phase, or imaginary, component *χ*″ = *χ* sin ϕ. The real *χ*′ has a plateau in the low frequency regime and equals the initial magnetic susceptibility from the DC magnetization curve without hysteresis. As the frequency of the alternating field increases, the magnetization of the particles is not able to keep up with the alternating field, and χ′ decreases. At the *T_B_*_,_ *χ*″ shows a maximum at the characteristic relaxation frequency, where a given angular frequency ω, is equivalent to *τ**_N_* = 2πω^−1^.

#### 2.1.2. Case Studies

During the last decades, magnetic nano-glass ceramics attracted great attention as promising candidates to a great diversity of applications, depending on their soft or hard characteristics [81]. Promising applications are emerging in areas such as adsorption, catalysis, ferrofluid technology, or magnetic resonance imaging (MRI) along with smart materials applications such as environmental, chemical, biomedical, and pharmaceutical sensors [82,83,84,85,86,87,88,89,90,91,92,93,94].

The need for improved soft magnets is continuously capturing the attention of researchers. The nanocrystalline and the amorphous materials are still being refined in order to increase M_s_ with the introduction of alloys that are more amenable to the fabrication of large-scale parts. Powder cores opened the door for nanoparticle-based composites, which can be produced with both top-down and bottom-up approaches.

As soft magnets, nano-glass ceramics are mostly based on cubic spinel ferrites, such as magnetite (Fe_3_O_4_) and zinc ferrites (ZnFe_2_O_4_). They can show ferromagnetism or ferrimagnetism combined with superparamagnetism exhibiting narrow hysteresis loops with small coercive fields, and their solid solutions can generate high amounts of heat. Hence, they can be found in biomedical applications, such hyperthermia and drug targeting, diagnostics applications, such magnetic resonance imaging (MRI), and separation/selection processes, but also in information storage systems, ferrofluid technology, and magnetocaloric refrigeration [81,95].

Two primary synthesis methodologies are currently used to produce inorganic soft magnetic glasses: melt-quenching and sol-gel. Although recent, sol-gel methodology allows more straightforward and more accurate control over the glass structure and/or the glass morphology, which is outstanding in the production of glasses with a well-defined magnetic response.

Ferreira da Silva et al. [96] used sol-gel to synthesize inorganic nanocomposites with ZnO-Fe_2_O_3_-SiO_2_ amorphous matrix dispersion of crystalline zinc ferrite NPs. The nanocomposite system exhibited ferro- or ferrimagnetic interactions and superparamagnetism with a blocking temperature of −259 °C. The annealing causes partial dissolution of the zinc ferrite NPs and precipitation of hematite NPs, in all the studied compositions.

Graça et al. [97] studied AC and DC conductivities in 88SiO_2_-6Li_2_O-6Nb_2_O_5_ (mole percent) sol-gel glasses heat-treated at different temperatures from 500 °C up to 800 °C. LiNbO_3_ crystals precipitated during heat treatments ~500 °C, SiO_2_, Li_2_Si_2_O_5,_ and Li_3_NbO_4_ at ~700 °C. σ_dc_ decreased with the concentration of the precipitated crystalline phases due to the decrease of charge carriers.

Ferreira da Silva et al. [98] synthesized MgFe_2_O_4_ spinels (where x varies from 1.25 to 10 mol% in the basic composition xFe_2_O_3_-5MgO-(95-x)SiO_2_) by sol-gel to be deposited over glass substrates. The effects of heat treatments at 500 °C and 1000 °C for 1 h were studied. For the amorphous compositions, heat treatments at lower temperatures revealed paramagnetic behavior associated with ferrimagnetic interactions inside the NPs and paramagnetic interactions between them. At the highest temperature, a combination of ferro- or ferrimagnetic behavior with superparamagnetism was revealed. The blocking temperature was below 60 K for samples with x ≤ 5 and around 160 K for the sample with x = 10.

Lithium ferrites exhibit great interest for applications in magnetic recordings, microwave systems, and computer memory chips. Nevertheless, their high volatility stays a major drawback. To overcome this obstacle, Graça et al. [99] proposed a sol-gel synthesis of 88SiO_2_-2Li_2_O-10Fe_2_O_3_ (percent mole). The sol-gel products were heat-treated at different temperatures, from 250 °C up to 1000 °C, for 4 h. At the lowest temperature, the sample presented a paramagnetic behavior, although an enhancement of ferrimagnetic behavior as temperatures raised was observed, reaching the highest value at 1000 °C.

Alternatively, Baikousi et al. [100] synthesized the CaO-SiO_2_-P_2_O_5_ glassy matrix by sol-gel and then homogeneously dispersed SPIONs within. This nanocomposite material showed both bioactive and magnetic performance as well as high stability towards crystallization, even at high temperatures (~800 °C).

Talaat et al. [101] proposed a new spinel composition, Fe_71.7_Si_11_B_13.4_Nb_3_Ni_0.9_, synthesized through the modified Taylor-Ulitovsky technique for hyperthermia cancer treatment. The first main goal was to overcome some of the SPIONs drawbacks such as high aggregation tendency, relatively low saturation magnetization, and moderate heating efficiency. This work successfully proved that using single or multi-microwires possessed considerable heat response to be used in cancer treatment. To address the same issue, Baino et al. [102] synthesized inorganic glass and glass-ceramic materials (on SiO_2_-CaO-Fe_2_O_3_ based-system) by sol-gel and evaluated their hyperthermia performance. Three compositions were studied—60SiO_2_-40CaO, 60SiO_2_-38CaO-2Fe_2_O_3_, and 60SiO_2_-30CaO-10Fe_2_O_3_. The sample with the high amount of Fe_2_O_3_ showed the best hyperthermia performance, while future biocompatibility studies will determine its commercial interest.

Ponsot et al. [103] incorporated waste, borosilicate glass residues, and iron-rich slags to produce glass-ceramics through the melting-quenching technique. Pollutants presented in the residues were stabilized in mixtures up to 75–50% from the glass and 25–50% from the iron-rich slags. Melting temperature was in the range of 900 °C up to 1000 °C. Furthermore, magnetite was formed during the melting process, exhibiting an intense heating when under a magnetic field. This result proved to be very interesting for hyperthermia treatments, since the product is biocompatible.

As hard magnetic materials, most of the nano-glass applications deal with M-type barium (BaFe_12_O_19_) [104,105] and strontium hexaferrites (SrFe_12_O_19_) [106,107] with small sizes, from 5 to 10 nm, to produce high-coercive permanent magnets of several kOe, since they can show large uniaxial magnetic and shape anisotropies, which prevail over the superparamagnetic behavior [108]. Such magnetic glasses are usually present in electric motors, ultrahigh-density magnetic recording media, approaching several TB cm^−2^, and magnetic separation processes [107,109].

More recently, works related to the magnetic properties of inorganic glasses are scarce, as is the search for its potential applications. Those applications usually demand soft magnetic materials, such as iron-based amorphous alloys. In order to fill this gap, our team is currently trying to expand knowledge on magnetic silica-titania flexible glasses obtained by sol-gel.

### 2.2. Solar Cells and Transparent Photovoltaic Devices

Solar photovoltaic (SPV) cells stand up as a critical player in the global renewable energy sector. SPV cell technology is a promising, clean, and sustainable energy source developed rapidly in recent years [110,111]. However, its efficiency loss may range as high as 25–30%, and a lifetime can be compromised.

Lately, transparent photovoltaic glasses emerge as the most cutting-edge new solar panel technology that promises to be a game-changer in expanding solar scope. These transparent solar panels (that resemble regular glass) can generate electricity from windows—in offices, homes, car sunroofs, or even smartphones. Some of these promising technologies are already in the advanced stages of development and could hit the market soon. Researchers at Michigan State University estimate these fully transparent solar panels’ efficiencies to be as high as 8% [112]. Their lower efficiency is bound to be overcompensated by their potential scale of deployment. Today, there are approximately 24 models of SPV technologies made from different materials and methods [113,114].

#### 2.2.1. Solar Photovoltaic (SPV) Cells

SPV cells are semiconductor devices that convert sunlight into electricity through the photovoltaic effect. The semiconductor material (as silicon) has the property to eject electrons after absorbing photons (*hν*) from sunlight (leaving holes that are filled by surrounding electrons). Then, SPV cell directs the electrons in one direction, forming an electrical current (Figure 6) that is proportional to the number of *hν* absorbed.

Until recently, the bunch of flexible materials disposed for electronic applications was limited to plastics, polymers, and composite membranes. Glass emerges as an unexpected flexible material opening the door for novel highly resilient flexible-electronic devices. Corning^®^ Willow^®^ Glass is a thin flexible glass already in the market, which first potential application is for solar energy production.

#### 2.2.2. Transparent Solar Photovoltaic (TSPV) Cells

A transparent solar photovoltaic cell (TSPV) is a new solar cell concept. Here, the absorbed *h*(*ν*) are also converted into electricity, while the visible range of the electromagnetic spectrum is transmitted through the glass cell. To implement this idea, two main strategies were developed: (i) visible light absorption, in which light in the visible region is partially absorbed and transmitted, and (ii) luminescent solar concentrator (LSC) technology (using luminescent materials), where the absorption range is moved from visible into UV or infrared (IR) range through LSC. LSC is composed of organic salts that are designed to absorb specific UV and/or IR light and to transform it into another wavelength out of visible range. This new wavelength is then guided to the edge of the window, being converted into electricity by thin SPV cell strips [116].

Between SPV and TSPV, there is plenty of room for intermediate solutions. The German manufacturer Heliatek Gmb, for example, developed a partially transparent SPV cell, which can absorb about 60% of the sunlight it receives [117]. Compared to the conventional SPV cells, the partially transparent solar panels have lower efficiency at 7.2%. However, solar power generation can be increased by adjusting the balance between the transmitted and absorbed sunlight.

#### 2.2.3. Case Studies

In the semiconductor market (remember SPV and TSPV are semiconductor devices), glass is making serious inroads (Table 4 and Table 5). Used either as a permanent or a temporary material within the semiconductor manufacturing processes, inorganic glass plays a key role in the semiconductor industry either as IR cut filter for complementary metal-oxide-semiconductors (CMOs), image sensor technology (CIS), microfluidics devices, actuators, and sensors. Nevertheless, its ultimate quest comes with flexible glass substrates for solar energy production and flexible glass packaging in electronic systems. In MEMs and electronics applications, glass wafers are used in wafer packaging of sensitive components due to their superior functionality and extreme reliability over time and in the face of harsh environments. As a carrier substrate, glass is chosen due to qualities such as thermal stability and chemical resistance, and, in both cases, mismatch needs to be avoided.

Peng et al. [118] used Corning^®^ Willow^®^ Glass substrate for (sequential sputtering) flexible thin films of Cu, Sn, Zn, and S. The deposition process occurred at room temperature. The films showed a good efficiency (in solar cells) in the horizontal position but a reduced efficiency when in bent positions of 50 mm radius (3.08–2.41% respectively). This works opens the possibility to apply flexible glasses effectively for solar energy. In another application for the same glass substrate, Sheehan et al. [119] produced a dye-sensitized solar cell (DSSC). The work showed a higher power conversion efficiency (7.42%) for the fluorine-coated glass when compared with two commercial flexible glasses, both based on tin oxide modified with indium and fluorine but with a different technique (screen-printing) and material (TiO_2_ electrodes). The result relies on the stability of this oxide properties (ohmic resistance and optical transmission) even after a heat treatment at 500 °C. Nevertheless, the Willow^®^ Glass, if coated with fluorine, should increase its efficiency, showing a promising outcome.

Despite SPV cell recognition, the degradation of their conversion efficiency overtime stays a significant drawback. Dust, grime, organic particulate matter, and other inorganic pollutant deposition, particularly in plateau geometries, either industrial or urban, contribute to the SPV cell’s lifetime reduction. Further, periodic cleaning of the SPV panels is often restricted by water/workforce budgets. The development of novel SPV cell surfaces through micro-and/or nano-engineered coatings opened a new approach towards the fabrication of self-cleaning panels’ covers. The top layer of silicon solar cells, the SPV cell with wider commercial use, is a cover glass having different functions [120]:(i)reducing the high reflection coefficient of silicon to improve cell efficiency;(ii)acting as a radiation barrier and optical coupling element;(iii)protection against debris and aggressive agents present in the air. Surface fouling, particularly in cities and when limited periodic cleaning is possible, remains an issue.

Several techniques were developed and implemented to achieve large-scale self-cleaning coatings on SPV cell systems (with extremely high wettability contact angle ~150°). Wet chemical methods include NPs coatings using a suitable binder, layer-by-layer assembly, and sol-gel processes. Sol-gel methodologies, namely, spray-coating, dip-coating, and aerosol-assisted vapor-deposition, stand out by their simplicity and high efficiency. Such coatings should help to keep the photovoltaic (PV) modules clean without compromising their light transparency. Most importantly, such self-cleaning glazing structures became very useful in diverse fields of solar energy application areas, such as solar radiation transmission, building integrated photovoltaics (BIPV), solar panels, and concentrated solar power (CSP) systems.

One of the authors developed a sol-gel-based, highly transparent, self-cleaning coating with tunable wetting property by synthesizing single component silane-modified base and acid-catalyzed silica sol [121]. A dip-coating approach was used. A static contact angle (WCA) as high as 150° and contact angle hysteresis (CAH) of ~2° was achieved with these coatings. Glass’s maximum transmission used in solar glass cover was found to increase from 91.8% to 95.5%, and concomitantly minimum reflectance was found to reduce from 8.7% to 3.2%. Such antireflection behavior was further investigated by conducting ellipsometry studies, where the refractive index of the coating was found to be 1.35 with film thickness 103.54 nm.

Lim et al. [122] studied the use of thin films, hydrogenated, amorphous silicon, and silicon-germanium cells to use as semi-transparent solar cells. The fabrication technique used for that purpose was radiofrequency (rf)-plasma-enhanced chemical vapor deposition working at 250 °C and 1.6 Pa. Their results proved that, related to transparency and efficiency of the cell, the transparent conductive oxides thickness should be superior to 300 nm. The combination of Si:Ge also showed better efficiency.

Nam et al. [123] explored a sacrificial layer approach to produce thin-film solar cells. Most of these layers require etching processes, strong acids that are not very specific and can damage metal electrodes. Therefore, they studied the application of a water-soluble sacrificial layer based on germanium oxide produced by dry oxidation at 510 °C for 45 min. This kind of sacrificial layer could also be used to manufacture other solar cell types.

On behalf of transparent solar cells, Sutha and collaborators [124] worked on the fabrication of a solar panel based on aluminum oxide coatings. They were able to reach levels of 95% transmittance. In another work at the same object of interest, Patel et al. [125] investigated the use of ZnO/NiO in a multi-functional transparent photoelectric device. Their solar cell demonstrated efficiency of 6% and a transmittance of 69.6%.

More recently, Wang and co-workers [126] developed a fully inorganic solar cell made with CsPbI_2_Br/CuBr_2_. Their device showed an efficiency of 16.15% and high stability, maintaining 95% after a month.

### 2.3. Photonic Crystals

A new class of optical materials known as photonic crystals (PCs), or photonic bandgap materials (PBGs), holds promise for transfer the full functionality of semiconductor devices into the optical field, combining high integration with high-speed processing and quantum computing [127,128,129,130]. Novel types of waveguides and optical fibers, new filters, high-speed switches, low-threshold micro-lasers, high-performance LEDs (light-emitting diode), photonic for VLSI (very large scale integration), along with smart materials applications such as environmental, security, energy, transport, biological, and chemical sensors are among the new opportunities in scientific and industrial areas such as information and communication, industrial manufacturing and quality, life sciences and health, emerging lighting, electronics and displays, security, metrology and sensors, design and manufacturing of components and systems, agriculture and food, and automotive and transport [3].

The growing demand for photonic crystal provided a significant boost to the global photonic crystal market as more people are shifting their preferences to this growing sector. The European photonics market is projected to grow at a CAGR of 8.4% leading up to 2022, being the critical enablers for the future mega-markets such as Internet of Things (loT), cybersecurity, quantum technologies, healthcare, and additive manufacturing, among others (Figure 7).

PBGs or PCs are structures whose refractive index, n, or dielectric constant, e, are periodic on a length scale of the order of optical wavelengths, which prevents light from propagating through the structure due to Bragg reflection [134]. Depending on the dimensionality of this periodicity, one can have PBGs in one, two, or three dimensions (1D, 2D, or 3D). Distributed Bragg reflectors (DBR), also called Bragg mirrors, are examples of 1D PBGs. Their overall reflectivity can be very high, even higher than that of metallic mirrors, and it is due to Bragg reflection. The construction of 3D PBGs, on the other hand, poses several difficulties. A wide range of techniques were used to fabricate 3D PBGs, sol-gel being the most promising ones.

#### Case Studies

The 1D PBGs consist of an alternating, high, and low refractive index dielectric multilayer stack, where the optical thickness, nx, of each layer equals k/4, k being the vacuum wavelength for which Bragg reflection occurs, n the refractive index of the materials, and x the layer thickness. An essential application of PBGs is made possible by the controlled introduction of defects, along which light may propagate within the stop band in a manner similar to the impurity levels of doped semiconductors. For a 1D structure, an example is the Fabry–Perot microcavity [135,136,137], which may be achieved by introducing an extra layer or the suppression of a layer within a multilayer stack. The cavity layer may also be doped with rare-earth ions [135], leading to exciting changes in their photoluminescence behavior. While spontaneous emission is inhibited within the stop band, where there are no photon modes available [134], within the cavity passband, the photoluminescence is enhanced by a factor of the order of the quality factor of the cavity.

Concerning 3D PCs, a variety of methods, such as gravity sedimentation [138,139,140,141], electrostatic repulsion [142,143,144,145,146,147], capillary forces induced convective self-assembly [148,149,150,151], and electric field-induced assembly [152,153,154,155], were developed. In this context, one of the main obstacles for putting into practice the interesting optical and structural properties of colloidal crystals actual devices is the incompatibility of the time-consuming and the unclean self-assembly crystallization techniques commonly used to make colloidal crystals with fast and dirt-free technology required to fabricate devices. A new approach to colloidal crystallization of submicrometer diameter spheres that overcomes some of the obstacles mentioned above was proposed by Jiang and McFarland [156]. In fact, it was shown that it is possible to obtain ordered colloidal structures by spin-coating technique, reducing the deposition time sensitively. Particularly, in these studies, the attention was focused on finding the best conditions (nanoparticles concentration—solution and velocity of spin rotation) to obtain large self-assembled areas using silica spheres.

Thus far, the 3D structures prepared by sol-gel are of artificial opal or inverse opal types. The process of colloidal crystallization was extensively studied by one of the authors, leading to the development of several methods to make colloidal crystals with fewer crystalline defects [136,156,157,158,159,160,161] and a smart material application such as flexible photonic crystal for strain sensing [53].

### 2.4. Smart Materials

#### 2.4.1. Smart Materials and Internet of Things (IoT)

Smart materials that can respond to external stimuli were explored in recent years, and once this market is continuously growing, new avenues and applications will start to open for them to be used. More recently, smart materials were inserted on the concept of Internet of Things. This concept is in close relation with the project and the idealization of smart cities, where everything is connected, providing to each individual the possibility to control, monitor, and manage devices remotely [162]. Not only that, IoT also finds important application further automating the line of production in industries such as: automotive, enabling the vehicles to be smarter and safer; healthcare, giving the possibility of accurate and quicker examinations and providing instant data to the doctors; retail and logistics, which can involve the use of less man power and consequently less man interaction; security, by creating even more precise and sensitive security systems; and agriculture, by developing more productive ways of treating and harvesting the crops.

The energy transduction principles that are employed for chemical and biological sensing involve radiant, electrical, mechanical, and thermal types of energy. Specific sensing concepts are further implemented with each energy transduction. Sensors based on transduction’s radiant energy can employ intensity, wavelength, polarization, phase, or time resolution detection. Sensors based on the electrical energy of transduction can employ conductometric, potentiometric, or amperometric detection. Sensors based on the mechanical energy of transduction can employ gravimetric or viscoelastic detection. Sensors based on thermal energy of transduction can employ calorimetric or pyroelectric detection [163].

#### 2.4.2. Case Studies

Water quality, water distribution, greenhouse gasses emission, and use of pesticides in agriculture are some of the topics guiding many efforts from scientists all over the globe. Almeida et al. [164] prepared an aluminum-silicate mesoporous glass (Si_1−x_Al_x_Na_x_O_2_ (0.1 < x < 0.33)) by using a simple and low temperature synthesis. This material showed a promising result related to pH dependence since no difference was observed in its effectiveness. However, even theoretically, this composition presents a higher capacity to make ion exchange than other commercial products used for comparison; practically, their sorption efficiency was much lower. They also proved that, with multiple exchanges and combustion cycles, this drawback is feasible to overcome. On the plus side, their material proved to have many reuse cycles until sodium content reaches its exhaustion. Moreover, the material can be customized for different applications in media depending on Na:Si ratio.

Silicon wafer fabrication produces glass substrates used in a variety of biotechnology applications. Borosilicate glass, a quality option in glass for medical devices, offers superior resistance to high levels of heat and energy as well as radiation exposure, such as within X-ray equipment.

Wafers are also used in microfluidic chip production using nanoimprint lithography, where glass acts as a substrate. Glass offers the clear optical transparency required in many biotechnology applications, making it a common choice for use as a capping layer over devices made from silicon. Wafer bonding processes such as anodic and thermal bonding create a hermetic seal.

Silica and titania glasses are often used to produce solutions for medical and biological issues thanks to their easy customization and inertness to the human body and their chemical and physical properties, which provide several different kinds of applications [165,166,167,168]. One of the most popular usages is as a bactericidal agent. Akhavan and Ghaderi [169] immobilized CuO nanoparticles in a silica film made by sol-gel technique at room temperature with dip-coating to be applied against *Escherichia coli* bacteria. On their tests, the greater the temperature was, the better was the inactivation of the bacteria at the surface of the film, demonstrating that Cu nanoparticles are much better photocatalysts than CuO nanoparticles. Additionally, all the results were better in the presence of light irradiation than in the dark.

One of the most explored potentials of silica and titania materials in the medical field is drug delivery, mainly because of their controlled release kinetics property. For that purpose, Bhattacharyya et al. [170] studied sol-gel silica films coating Ti roads and wires with two drugs: farnesol and vancomycin to treat methicillin-resistant *Staphylococcus aureus*, a dangerous and resilient bacterium most found in implants. Their results proved a fair use of the films for titanium implants once they had the capacity to associate with the medicaments and release them entirely in five days. Combining both gave the best outcome by killing almost 100% of the bacterium.

Our team studied the use of amorphous titania NPs as a new route for cancer treatment by using photo or sono-dynamics therapy, and the results showed a promising application of the synthesized NPs due to their response to ultrasound stimulation [60,171].

Related to implants, Catauro et al. [171] used sol-gel and dip-coating to evaluate improvements on bioactivity and compatibility of SiO_2_ and CaO films coating Ti plaques. Many different proportions of SiO_2_ and CaO were applied in the tests with simulated body fluid, and the best results were obtained for 0.3 and 0.4 CaO, with 0.7 and 0.6 of SiO_2_, respectively. Nevertheless, coated Ti plaques were more biocompatible and activable than uncoated ones, promoting the use of coating for these applications in the medical field.

Hydroxyapatite is an important material for implants but can be also used as sorption material, and by mixing it with titania, for example, it can potentially enhance photocatalysis, degradation of pollutants, and inhibit bacteria activity. Kaviyarasu et al. [172] synthesized a hydroxyapatite and TiO_2_ composite (3:1) by the sol-gel method at room temperature. The tests used Rhodamine-B related to photocatalysis and degradation for anti-microbial test *Bacillus* spp. and *E. coli*. According to their results, this device had an excellent growing inhibition for both bacteria.

Thermistors are devices with resistance that depends on temperature. They can be positive or negative, meaning that resistance grows with higher temperature (positive) or decreases with higher temperature (negative). Such devices have many applications, such as industrial, medical, and energy related. One of the gaps is that they can be expensive and fragile. Sohal et al. [173] worked in a tin oxide thermistor device production using a low-cost precipitation method at room temperature. They proved with thermal resistance tests that their devices can be used as human body temperature monitoring devices.

Amperometric sensors are susceptible devices used to detect trace elements in different fields, such as medical diagnosis, food safety, and environmental monitoring. Fan et al. [174] manifested an easy way of producing these sensors over Willow^®^ Glass using Ag/AgCl and carbon graphite inks to detect hydrogen peroxide. Their results showed a very linear response of both sensors being more rapid and sensitive. They also tested the capacity after bending stress, and after 100 times, both lost 50% of the signal intensity. These results are promising but still require much work and effort to extend their usage and permit more customization of their application without prejudice in the device response.

## 3. Prospective Areas and Closing Remarks

An interesting and growing area is related to electronics implants. For that matter, Melzer et al. [175], Choi et al. [176], Bermúdez et al. [177,178], Ota et al. [179], and Hua et al. [180] produced high-end electronics implants using polyethylene terephthalate (PET), MoS_2_-graphene, polyimide layers, polydimethylsiloxane (PDMS) coated glass slides, polyethylene naphthalate (PEN), and Kapton HN film (DuPont), respectively, as supports for their materials. Solely one among these recent researchers used glass as support for the electronic implant, which indicates that glass still fails to be a plausible choice. One of the main reasons for that is the obstacle of having highly foldable glass practically and easily. Moreover, obtaining thin films with a smooth and continuous surface is not always possible, as such kinds of devices demand. Notwithstanding, we saw eagerness and growth, as reported early in this review article, of foldable glass being used in the phone market, virtual reality devices, and even cars with an automatic pilot. Hence, it is expected that we could have an accession of the use of glass and glass-ceramic materials in this field in the future.

Other great fields of inorganic flexible glass application that are yet to be explored are those of field effect transistors (FET) and organic field effect transistors (OFET). There are several papers published in this area using graphene, polymers, and even cellulose to give flexible characteristics to the devices [181,182,183,184,185,186,187,188,189,190,191,192,193,194,195,196]. Among all, only one, Zocco et. al., reported the use of Willow^®^ Glass and compared the efficiency between the inorganic glass substrates with the flexible paper [196].

Encapsulating materials is a very interesting area that inorganic flexible glass could also enter due to the characteristics discussed in this paper. This field is important with the contribution of polymers and graphene [197,198,199,200,201,202,203,204,205,206,207], although inorganic flexible glass materials also permeated the area with Willow^®^ Glass and thin glass/polymer composition using roll-to-roll processing [115,208,209,210]. However, there is still plenty of room for the inorganic glass influence to grow.

Other possible substrates, such as plastic substrate and stretchable elastomers, are recently appearing as alternatives to glass in the production of flexible displays and stretchable electronic devices. The performance of their displays are being compared taking into account parameters such as surface quality, transmission, and thermal and dimensional stability. These soft substrates are usually natural rubber (NR), styrene butadiene rubber (SBR), ethylene-propylene-diene monomer (EPDM), polyurethane (PU), thermoplastic polyurethane (TPU), predominant poly(dimethylsiloxane) (PDMS), etc., which can reversibly endure high deformations (>200%) [211]. However, the use of such materials often results in low electrical mobility and high electrical resistivity of electronic devices and mainly to a reduction in the working temperature.

Devices with good flexibility or stretchability based on a silicon membrane, single-walled carbon nanotubes, or poly(ethylene naphthalate) film were prepared and fabricated by standard methods on a carrier substrate such as a Si wafer or a glass plate [212] or even directly on the flexible/elastic substrate, including low-temperature deposition, solution processing, nano-/micromolding, and electrospinning. Currently being explored as an emerging technology is the use of transfer printing in the fabrication process of flexible and stretchable electronic devices. One example is the integration of small crystalline-silicon circuits (chiplets) in the active-matrix organic light-emitting-diode (OLED) displays [201,213]. Studies showed that these chiplets could be transfer-printed on the glass substrate via an elastomeric stamp to help build the integrated circuits. The OLED display is then formed and connected to the chiplets, demonstrating an exceptional performance and highlighting the effectiveness of the transfer printing method.

Another example that compares the properties of different substrate materials deals with OLEDs and amorphous Si thin film (α-Si) transistors (TFTs) on both flexible glass and thin stainless steel sheets. These studies revealed that the yield of OLEDs on stainless steel foil substrates was lower than that on glass because of the surface roughness of stainless steel [214].

The cost involved in manufacturing such smart materials is of utmost importance for developing and realizing practical applications. Therefore, the design of different materials and devices and the fabrication processes for implementing such a strategy remain challenges.

From ancient Egypt to AM technology, from simply hollow glass to foldable or multi-layered morphologies, inorganic glass became an unavoidable material in opto-electronic devices and smart materials applications. After millennia of compositional and technological developments challenging new applications and/or structures, there is plenty of room for improvement. Hopefully, this review inspires researchers and specialists to develop new applications of this outstanding material, leading to innovation.

## Figures and Tables

**Figure 1 materials-14-02926-f001:**
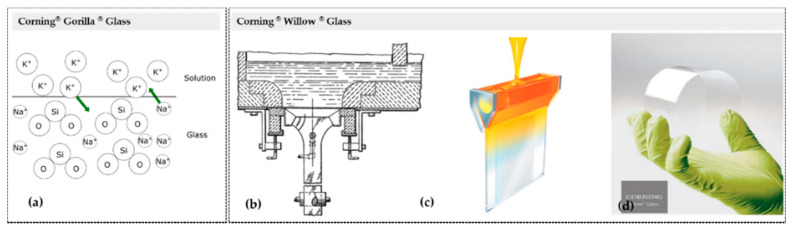
Corning^®^ keeps leading innovation in high-tech glass materials: (**a**) Corning^®^ Gorilla^®^ Glass fabrication technic; (**b**–**d**) Corning^®^ Willow^®^ Glass, (**b**)—down-draw fabrication process, (**c**)—overflow fabrication process, and (**d**)—Flexible Corning^®^ Willow^®^ Glass product. ((**b**)—reprinted from [14] © (2010) with permission from Wiley; (**c**)—reprinted (adapted) with permission from [15] © (2016) American Chemical Society; (**d**)—reprinted from [16] © 2021 with permission from Wiley).

**Figure 2 materials-14-02926-f002:**
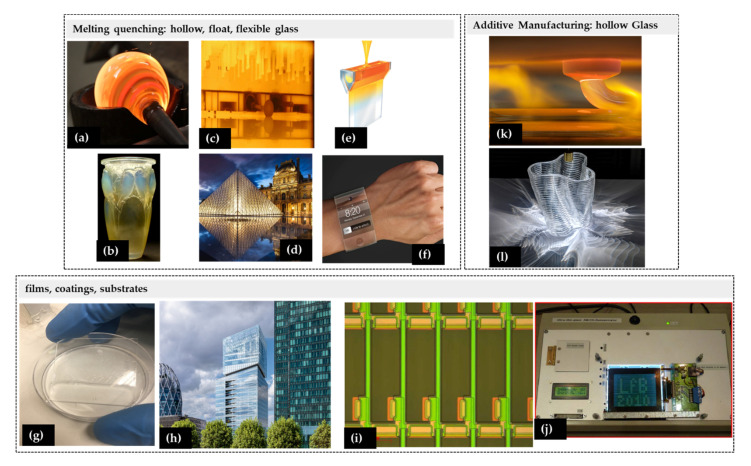
Examples of different glass fabrication methodologies. Melting-quenching: (**a**) glassblowing; (**b**) Lalique glass; (**c**) float chamber, credits by Saint-Gobain^®^ Glass Portugal; (**d**) Louvre Museum entrance, Saint-Gobain^®^; (**e**) overflow fabrication process (reprinted (adapted) with permission from [15] © (2016) American Chemical Society); (**f**) Apple^®^ Willow^®^ glass watch; coatings and substrates; (**g**) SiO_2_/TiO_2_ Sol-gel coating on a standard glass substrate; (**h**) coated glass, Saint-Gobain, Paris, Valode & Pistre Architects, credits by Saint-Gobain^®^ Glass Portugal; (**i**,**j**) TFT backplane fabricated on flexible glass and flexible glass AMLCD prototype (reprinted from [75] © 2021 with permission from IEEE; Additive Manufacturing—Hollow Glass); (**k**) glass 3-D printing process by Steven Keating, MIT CC BY-NC-ND 3.0 available at [74], (**l**) by Andy Ryan, MIT CC BY-NC-ND 3.0 available at [74].

**Figure 3 materials-14-02926-f003:**
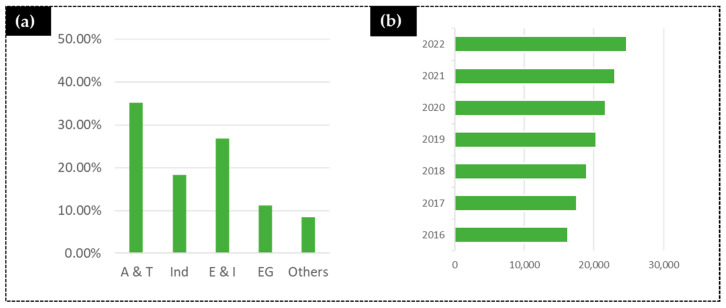
Soft magnetic materials (**a**) world market share (2016). A&T—automotive and transportation; Ind—industrial; E & I—electronics and instrumentation; EG—energy generation. (**b**) Market transactions and predictions (in USD billion) (based on images from [77]).

**Figure 4 materials-14-02926-f004:**
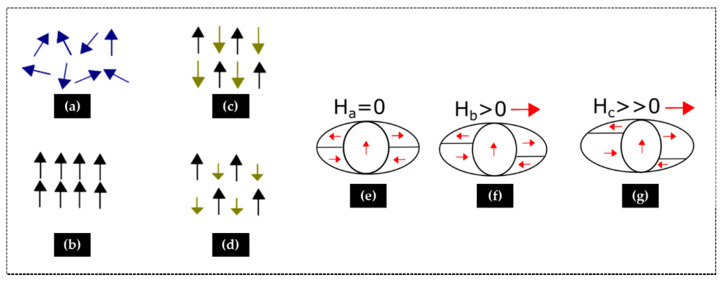
Magnetic interactions: (**a**) paramagnetic, (**b**) ferromagnetic, (**c**) antiferromagnetic, and (**d**) ferrimagnetic. Scheme of domains growing in a ferromagnet at: (**e**) zero magnetic field; (**f**) weak magnetic field; (**g**) strong magnetic field.

**Figure 5 materials-14-02926-f005:**
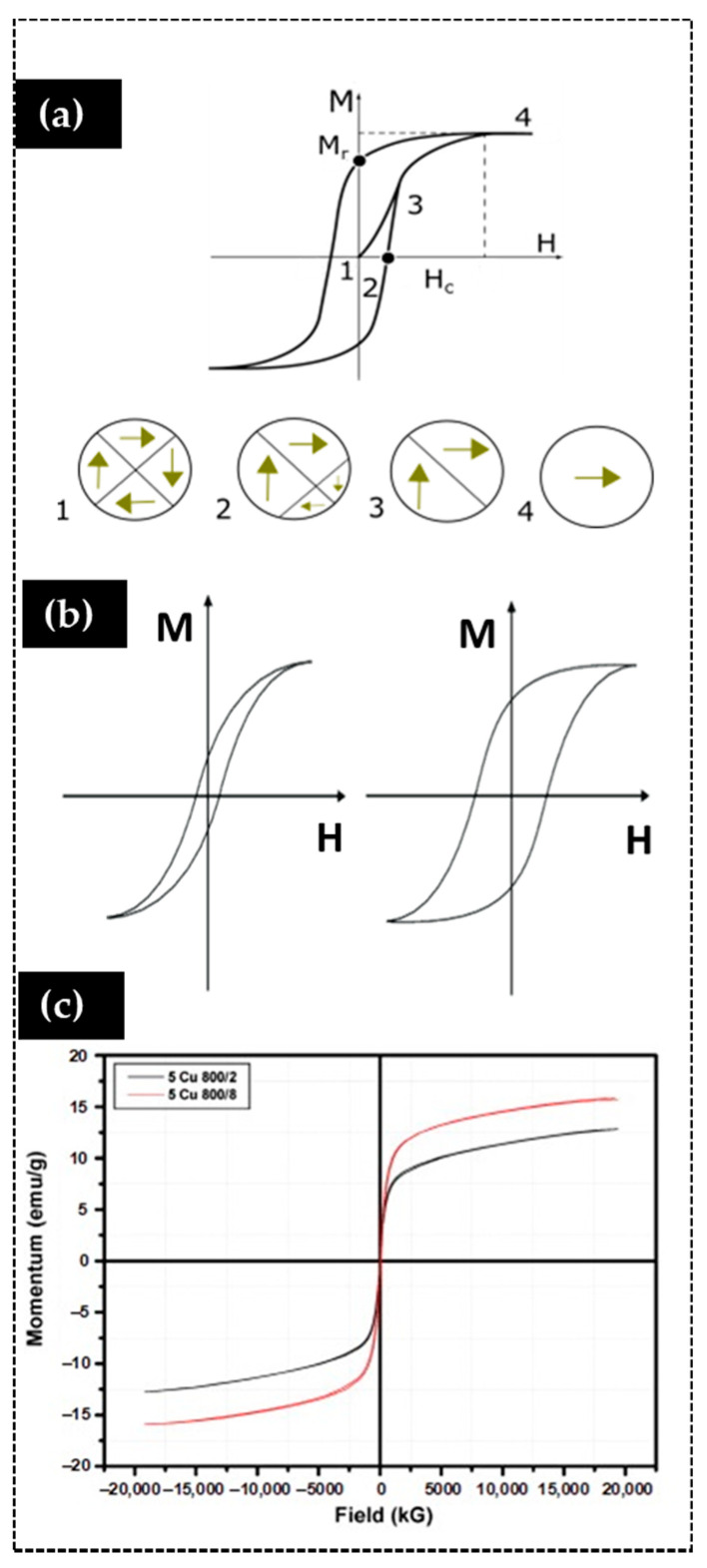
Hysteresis loop and domain growth of: (**a**) ferromagnet; (**b**) soft and hard magnetic materials hysteresis loop, respectively; (**c**) magnetic field dependence of the magnetization of a superparamagnet (reprinted from [78] ©2011 with permission from Elsevier).

**Figure 6 materials-14-02926-f006:**
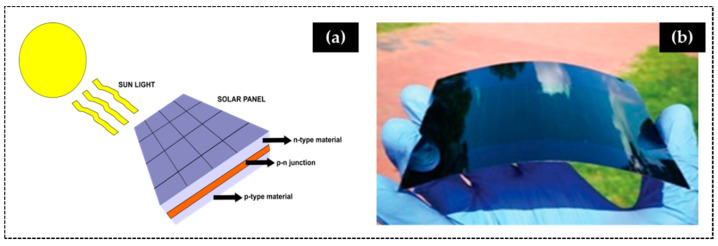
(**a**) Schematic of operation of a solar panel and (**b**) a solar cell produced with Corning^®^ Willow^®^ Glass (reprinted from [115] © 2021 with permission from Willey).

**Figure 7 materials-14-02926-f007:**
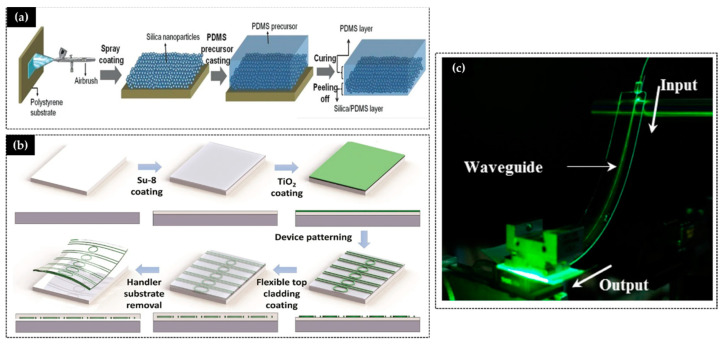
Examples of flexible glass for photonics applications: (**a**) schematic fabrication of a smart window (reprinted from [131] © 2021 with permission from Wiley), (**b**) schematic of the fabrication and de-attachment process of healthcare device (available at [132] printed with permission CC BY 4.0), (**c**) waveguides of photonics for telecommunications application (reprinted with permission from [133] © The Optical Society).

**Table 1 materials-14-02926-t001:** Breakthrough in glass melting-quenching technology (first generation on glass fabrication).

**Period**	**Native Glass ^1^**	**Applications**	**Ref.**
Pre-history	Natural glass: obsidian (volcanic glass) meteorite impact glass	Obsidian used in point of lances, arrow tips, and artwork.	[10,17,18,19,20]
**Period**	**Vitreous Slags**	**Applications**	**Ref.**
Ancient Egypt 12,000 BC	Glass glaze	Copper opaque and blue-tinted glass glaze used in ceramic, metal pieces, and natural rocks. Associated with high-temperature manufactories—such as ceramics and metallurgy—glass production would have emerged accidentally; melting copper minerals caused opaque and blue-tinted vitreous slag to form, and heating the ceramic pieces caused them to vitrify.	[10,17,18,19,20]
Ancient Egypt 7000 BC	Glass beads	Pyrotechnic experimentations with silica-clay mixtures eventually led to the creation of the glass beads.	[10,17,18,19,20]
Ancient Egypt 3000 BC	Glass glaze Glass bottle	Core-forming technique: bottles (and pots) were formed by winding glass ribbons around a mold of compacted sand. After cooling the glass, the sand was scraped from inside the bottle, leaving a hollow container with rough, translucent walls and usually lopsided shapes. A second glass manufacturing method utilizes molten glass poured in successive layers into clay or sand forms, thus creating a vessel of proper cohesion. The temperature needed to melt the raw materials was one of the main obstacles to glass melting/quenching technology.	[10,17,18,19,20]
Babylon 700 BC	Glaze, glass bottle Mosaic glass (millefiori)	Tin opaque and colored glass glazes used in ceramic pieces. ex: Ishtar Gate.	[10,17,18,19,20]
Han dynasty, China 206–200 BC	Glaze, glass bottle	Lead rich glass glaze (with a low melting point) used in ceramic pieces.	[10,17,18,19,20]
**Period**	**First Technological Revolution**	**Applications**	**Ref.**
Babylon 3000 BC Syria 3000 BC	Mesopotamian glassblowing First written glass protocol Glass large scale production Syrian glassblowing	Glassblowing utilized an iron blow metallic tube (about one and one and a half meters long). Bronze would not serve this purpose due to its lower solidus temperature (~850 °C versus 1538 °C for Fe), thus glassblowing had to await the iron age for its realization. The quality of glassblowing improved dramatically, and glass-drinking vessels became popular. Colored glass came into common use, with techniques for the production of many colors regarded as family secrets to be passed on from generation to generation of artisan’s families.	[10,17,18,19,20]
Alexandria 3000 BC	Alexandrian art of millefiori	Millefiori technique involved the production of glass canes or rods, known as murine, with multicolored patterns, which were viewable only from the cut ends of the cane. A murine rod was heated in a furnace and pulled until thin while still maintaining the cross-section’s design. It was then cut into beads or discs when cooled.	[10,17,18,19,20]
Rome Empire	Roman flat glass	Flat Glass was used to build high standard buildings for floors and wall decorations, but it was its use for windows, replacing mica and shells, where it contributed most to architecture (e.g., ruined cities of Pompeii and Herculaneum exhibited numerous traces of sheets of glass, probably used in the windows of the public baths). Imperial Rome’s fall and the instability caused by the Huns in medieval Europe caused the glass-producing centers to decline.	[10,17,18,19,20]
Europe Middle Age	Stained-glass windows	During the European Middle Ages, small glassmaking centers were established hidden in forests. This is the reason why sodium carbonate, a traditional glass modifier, was replaced by potassium carbonate during this period. The combination of the discovery of many new colorants (organic origin) with the glassblowing eventually led to the magnificent stained-glass windows of so many of the great cathedrals of Europe and the Near East.	[10,17,18,19,20]
Venice 8th Century	Secretes of glass manufacturing methods and techniques	The revival of trade with the Byzantine Empire led to renewed glass production in Venice. Under the pretext of protecting Venice from fire, Venetian artisans were forced to reinstall their kilns in Murano, where they would remain prisoners of their art.	[10,17,18,19,20]
**Period**	**19th Industrial Revolution ^2^**	**Applications**	**Ref.**
Europe 19th–20th Century (third quarter of the 19th through the first quarter of the 20th centuries)	Automatic hollow glass production	Blow, blow-and-blow, puff-and-blow, and press-and-blow processes. Lynch-10, delivering 25–80 pieces/min with bottle weights of 30–600 g. Roirant-R-7, delivering 20–80 pieces/min with bottle weights of 100–1200 g. Hartford-IS-12, delivering 10 pieces/min.	[10,17,18,19,20]
	Ceramic glass		
**Period**	**Second Technological Revolution**	Industrial Revolution allowed the production of sheets of glass large enough to allow more extensive use in architecture.	
Europe 20th Century USA 21st century	Pilkington float glass production Corning^®^ Willow^®^ Glass Corning^®^ Gorilla^®^ Glass	Industrial Revolution allowed the production of sheets of glass large enough to allow more extensive use in architecture. In the float glass process, molten glass flowed from the melting tank into a bath of molten tin, 3–4 m wide, 50 m long, and about 6 cm deep. The glass surface flowed and smoothe itself while on the float bath, taking on the outstanding surface quality of molten tin and thus requiring no further polishing. Tin was used as a flotation medium due to its surface tension (about 0.55 N/m), density (5.9 g/cm^3^ compared with 2.2–2.5 g/cm^3^ for glass), and low melting point (505 K). A new generation of glass—transparent, ultra-thin, flexible glass. A new generation of glass—ultra-thin, ultra-mechanical resistance glass.	[13]

^1^ Glass was used even before it was manufactured. Some of the natural phenomena that produced glass include the melting of magma and meteorite impacts, followed by rapid cooling. Natural glass was used for millennia as a raw material to produce works of art along with functional objects, such as the point of a lance or as arrow tips, where it competes with silex. Of all natural glass, obsidian was the most used due to its relative abundance. ^2^ Until the 19th century Industrial Revolution, glass was a luxury material produced for an aristocratic, royal, or priestly market. Most of the pieces were found in temples, palaces, or tombs rather than private homes.

**Table 2 materials-14-02926-t002:** Breakthrough in sol-gel glass methodologies adapted from [24] (third generation on glass fabrication).

Year	Sol-Gel	Applications	Ref.
1640	Van Helmont, Newman	Water glass, sodium silicate suspension by melting sand with excess alkali; on acidification, silica gel is obtained	[29,30]
1845	Ebelmen	Transparent glass following atmospheric exposure of a silane obtained from SiCl_4_ and ethanol	[31,32]
1913	Patrick	Quick and cheap method of making silica gel in large quantities	[33]
1931	Kistler	Aerogel by supercritical drying	[34]
1939	Geffcken and Berger	Single sol-gel method for preparing single oxide coating	[35]
1968	Nicolaon and Teichner	Alkoxide route to aerogels	[36]
1968	Stöber	Alkoxide route to NPs	[37]
1984	Avnir	Doped sol-gel materials	[38]
1985	Schott	Antireflective Amiran^®^ glasses	[39]
1985	Schmidt	ORganic MOdified SILicates (ORMOSIL)	[40]
1989	Carturan	Immobilization of living organisms within silica	[41,42]
1990	Avnir	Immobilization of enzymes within silica	[43]
1994	Brinker, Prakash	ORMOSIL aerogels bypassing supercritical drying	[44,45]
1998	Bright	First low-cost O_2_ optical sensor based on Ru-doped luminescent material	[46]
1998	Toshiba	Sol-gel optical coatings for TV screens	[47]
1999	Ozin	Periodic mesoporous organosilicas	[48,49]
1999	De Vos and Verweij	Long-lasting ORMOSIL-based membranes	[50]
2000	Hench	Sol-gel derived bioglass as third-generation tissue regeneration materials	[51]
2003	Cabot Corp.	Production of silica aerogels under ambient conditions	[52]
2007	Gonçalves et al.	Sol-gel photonic crystals (direct, infiltrated, inverse opals) Flexible photonic crystals for strain sensing	[53]
2012	Warren and Wiesner	Silica gels doped with high amounts of metal NPs of unprecedented conductivity	[54]
2012	Gonçalves	Hollow or dense silica/titania NPs for biomedical applications	[55]
2015	Gonçalves	Silica/ORMOSIL/superparamagnetic iron oxide nanoparticles (SPIONs) as efficient nuclear magnetic resonance (NMR) contrast agent	[56]
2018	Gonçalves	Sol-gel monophasic hybrid silica/titania-cellulose acetate membranes	[57,58,59]
2020	Gonçalves, Cauda	Titania as US sensitizer in cancer treatment	[60]

**Table 3 materials-14-02926-t003:** Breakthrough in AM glass technologies (fourth generation on glass fabrication).

Year	3D Printing	Applications	Ref.
2015	Fateri	3D printing through selective laser sintering (SLS) to white, porous (yet translucent) glass components	[61]
2011	Marchelli	3D inkjet printing of glass powders to white, porous (yet translucent) glass components	[62]
2015	Klein	First AM approaches to produce inorganic glass are based on fused deposition modeling standard for printing), although processes result in coarse glass structures and demand special expensive printing equipment due to high-temperature processing	[63]
2014	Luo	On laser beam molten glass fibers (processes result in coarse glass structures and demand special expensive printing equipment due to high-temperature processing)	[64]
2016	Kotz	Low-temperature glass printing methods come up. The first low-temperature glass printing process uses silica nanocomposite inks (to be cured by ultra-violet (UV) light) through stereolithography.	[65,66,67]
2017	Nguyen, Cooperstein, Destino and Dudukovik	A second approach used colloidal silica suspensions as ink writing material	[68,69,70,71,72]
2020	Sasan	The third one developed stereolithography of photocurable sol-gel precursors.	[73]
2020	Massachusetts Institute of Technology (MIT)	Developed by a team comprising the MIT Media Lab’s Mediated Matter Group, the MIT mechanical engineering department, the MIT Glass Lab, and the Wyss Institute at Havard University. The additive manufacturing platform G3DP° (glass 3D printing) can print glass in a variety of shapes, profiles, and colors and subsequently different optical properties and degrees of opacity (https://www.architectmagazine.com/ (accessed on 30 April 2021)).	[74]

**Table 4 materials-14-02926-t004:** Industrial directories for high-end materials in different applications.

**(i) Magnetic Glass Materials**	**Website (accessed on 30 April 2021)**
Tyndall National Institute	https://www.tyndall.ie/
Hitachi Ltd.	https://www.hitachi.eu/pt-pt
IBM Corporation	https://www.ibm.com/pt-en
**(ii) Solar Cells and Transparent Photovoltaic Devices**	**website**
ENF Solar—Solar Companies and Products	https://www.enfsolar.com
Solar Energy Directory	https://dir.list.solar/
Standford Energy Corporate Affiliates	https://seca.stanford.edu/research/solar-research-directory
Clean Energy Authority	https://www.cleanenergyauthority.com/
European Directory of Sustainable and Energy Efficient Building 1999 Eds. 2013	ISBN 978-1-873936-93-I (pbk)
ITRS-International Technology Roadmap for Semiconductors	http://www.itrs2.net/
International Roadmap for Devices and Systems (IRDS™) 2020 Edition	https://irds.ieee.org/editions
CompInfo—The Computer Information Centre	https://www.compinfo-center.com/
BoogarLists|Directory of Semiconductor Manufacturers	BoogarLists/Directory of Semiconductor Manufacturers tml
Global Semiconductor Glass Wafer Market 2018–2022 (Report)	https://www.researchandmarkets.com/research/kxqdv4/global?w=4
Glass Substrate in Semiconductor Market—Global Industry Analysis, Size, Share, Growth, Trends, and Forecast 2017—2025 (Report)	https://www.transparencymarketresearch.com/glass-substrate-semiconductor-market.html
What is driving the Growth of the Glass Material Market in Semiconductor Manufacturing? By Yole Development	https://www.i-micronews.com/products/biomems-market-and-technology-2020/
Solar Energy Directory	https://dir.list.solar/c/solar-panels/thin-film/flexible/
Texas Instruments	https://www.ti.com/
Murata	https://www.murata.com/en-us
Metrix	http://www.metrixvibration.com/
Dytran Instruments	https://www.dytran.com/
Wika	https://www.wika.com/en-en/startpage.WIKA
**(iii) Photonic Crystals**	**website**
Europe’s Age of Light (Roadmap)	https://www.photonics21.org/download/ppp-services/photonics-downloads/Europes-age-of-light-Photonics-Roadmap-C1.pdf
Towards 2020 -Photonics Driving Economic Growth in Europe. Multiannual Strategy Roadmap 2014-2020	https://www.photonics21.org/download/about-us/photonics-ppp/photonics-roadmap.pdf?m=1513605711&
European Photonics Industry Consortium (EPIC)	https://www.epic-assoc.com/database/
Synopsys	https://www.synopsys.com
Washington Information Directory	ISNB 978-1-5443-0075-7 ISSN 0887-8064
vlcphotonics	https://www.vlcphotonics.com
Synopsys	https://www.synopsys.com
Holland NanoRoasmap	https://www.hollandhightech.nl/sites/www.hollandhightech.nl/files/inline-files/Roadmap-Nanotechnology-HTSM-March-2018.pdf
SPIE	https://spie.org

**Table 5 materials-14-02926-t005:** Leading key players of the global high-tech glass market.

Company	Country	Website (accessed on 30 April 2021)
Schott Glaswerke AG	Germany	www.schott.com
Heliatek Gmb	Germany	https://www.heliatek.com/de/
Physee	European	https://www.physee.eu/
Corning	USA	www.corning.com
Crystran Ltd.	USA	www.crystal-gmbh.com
Sterling Precision Optics	USA	http://www.sterlingprecision.com
Ohara Corporation	USA	https://www.oharacorp.com
Precision Optical Inc	USA	https://www.precisionoptical.com
Tesla	USA	https://www.tesla.com
Ubiquitous Energy	USA	https://ubiquitous.energy/
Brite Solar	USA	https://www.sungoldsolar.com/
OAG Werk Optik	Ukraine	http://ritmindustry.com/
Nikon Corporation	Japan	www.nikon.com
Edmund Optics	Japan	https://www.edmundoptics.jp
Sumita Optical Glass	Japan	https://www.sumita-opt.co.jp
Hoya Corporation	Japan	https://www.hoya.jp
China South Industries Group Corporation Glass (CDGM)	China	http://cdgmglass.com
Hubei New Huaguang	China	http://www.hbnhg.com
Changchun Boxin Photoelectric Co	China	http://www.bxoptic.com/

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
