# Peer review of "What Is Driving the Growth of Inorganic Glass in Smart Materials and Opto-Electronic Devices?"

_materials, 2021, doi:10.3390/ma14112926_

Round 1

Reviewer 1 Report

Barcelos et al., in the sumbitted review, discuss inorganic glasses, mainly towards their applications in the field of magnetic glasses, photovoltaics, photonic crystals, and sensing materials. Inorganic glasses do hold an important place in various applications and the submitted review is quite comprehensive which also provides enough background to a broad audience. Owing to the breadth of the applications and bases covered, I recommend publication of the article after addressing below concerns:

  1. Table 5: Corning is headquartered in USA not the UK. The authors should correct the country of Corning.
  2. Line 484: The reference# 112 is an article from the University of Michigan, not Michigan State University. Please correct the statement.
  3. The references do not follow the MDPI referencing guidelines. The authors should correct the format of the references.
  4. The page numbers are not in a proper sequence. This may be a formatting issue which should also be addressed.

Author Response

Response letter

We did appreciate the Editorial Office and reviewers’ comments on the manuscript. We have revised the paper according to those suggestions and recommendations, which we do believe have improved the quality of the manuscript.

We believe we have addressed all the reviewer’s comments. The changes made have made in Track Changes mode, according to the Journal Guidelines.

We believe the manuscript has reached the Materials’ high publication standards, so we submit the manuscript to your final approval.

Sincerely yours,

MClara Gonçalves

#Reviewer 1

Comments and Suggestions for Authors

Barcelos et al., in the sumbitted review, discuss inorganic glasses, mainly towards their applications in the field of magnetic glasses, photovoltaics, photonic crystals, and sensing materials. Inorganic glasses do hold an important place in various applications and the submitted review is quite comprehensive which also provides enough background to a broad audience. Owing to the breadth of the applications and bases covered, I recommend publication of the article after addressing below concerns:

Table 5: Corning is headquartered in USA not the UK. The authors should correct the country of Corning.

Thanks for this indication, the alteration has been made.

Line 484: The reference# 112 is an article from the University of Michigan, not Michigan State University. Please correct the statement.

In the article the reference #112 was University of Michigan already.

Transparent solar panels for windows hit record 8% efficiency | University of Michigan News, (n.d.). https://news.umich.edu/transparent-solar-panels-for-windows-hit-record-8-efficiency/ (accessed February 19, 2021).

The references do not follow the MDPI referencing guidelines. The authors should correct the format of the references.

Thanks for this indication, the alteration has been made.

The page numbers are not in a proper sequence. This may be a formatting issue which should also be addressed.

Thanks for this indication, the alteration has been made.

Reviewer 2 Report

The historic development of inorganic glass and their recent applications for smart materials and opto-electronic devices have been reviewed in this article. Four applications, including magnetic glass materials, solar cells, photonic crystals, and sensing materials have been summarized and case studies were sampled. This article can be published in Materials after addressing the following questions.

  1. Many opto-electronic applications have been covered in this review. What are the smart materials applications? Which kinds of applications can be classified as smart materials? More information on this part should be added.
  2. Some typos in the text. Willey in page3,line98, should be Wiley. Marked in page3, line 140 may be market.
  3. One of the main topics of this review article is why the inorganic glass is still promising. Therefore, the comparison of inorganic glass with other transparent materials, such as organic glass, transparent plastic, transparent papers, etc. should be carefully detailed.

Author Response

Response letter

We did appreciate the Editorial Office and reviewers’ comments on the manuscript. We have revised the paper according to those suggestions and recommendations, which we do believe have improved the quality of the manuscript.

We believe we have addressed all the reviewer’s comments. The changes made have made in Track Changes mode, according to the Journal Guidelines.

We believe the manuscript has reached the Materials’ high publication standards, so we submit the manuscript to your final approval.

Sincerely yours,

MClara Gonçalves

#Reviewer 2

Comments and Suggestions for Authors

The historic development of inorganic glass and their recent applications for smart materials and opto-electronic devices have been reviewed in this article. Four applications, including magnetic glass materials, solar cells, photonic crystals, and sensing materials have been summarized and case studies were sampled. This article can be published in Materials after addressing the following questions.

Many opto-electronic applications have been covered in this review. What are the smart materials applications? Which kinds of applications can be classified as smart materials? More information on this part should be added.

We agree with the referee. So, in we highlighted the smart applications already present in the paper by adding some new sentences and by changing the name of the last class of examples to iv) Smart Materials.  

Abstract

Abstract: Inorganic glass is a transparent functional material and one of the few materials that keeps leading innovation. In the last decades, inorganic glass has been integrated into opto-electronic devices like optical fibers, semiconductors, solar cells, transparent photovoltaic devices, or photonic crystals and smart materials applications such as environmental, pharmaceutical, and medical sensors, reinforcing its influence as an essential material and providing potential growth opportunities for the market. Moreover, inorganic glass is the only material that is 100% recyclable and can incorporate other industrial offscourings and/or residues to be used as raw materials. Over time, inorganic glass has experienced an extensive range of fabrication techniques, from traditional melting-quenching (with an immense diversity of protocols) to chemical vapor deposition (CVD), physical vapour deposition (PVD), and wet chemistry routes as sol-gel and solvothermal processes. Additive manufacturing (AM) was recently added to the list. Bulks (3D), thin/thick films (2D), flexible Glass (2D), powders (2D), fibers (1D), and nanoparticles (NPs) (0D) are examples of possible inorganic glass architectures able to integrate smart materials and opto-electronic devices, leading to added-value products in a wide range of markets. In this review, selected examples of inorganic glasses in areas such as: i) Magnetic Glass Materials, ii) Solar Cells and Transparent Photovoltaic Devices, iii) Photonic Crystal, and iv) Smart Materials are presented and discussed.

And later…p. 6

During the last decades magnetic nano-glass ceramics attracted great attention as promising candidates to a great diversity of applications, depending on their soft or hard characteristics [81]. Promising applications are emerging in areas such as adsorption, catalysis, ferrofluid technology, or magnetic resonance imaging (MRI) along with smart materials applications like environmental, chemical, biomedical, and pharmaceutical sensors [82–94].

And even later p. 11

A new class of optical materials known as photonic crystals (PCs), or photonic bandgap materials (PBGs), holds promise for transfer the full functionality of semiconductor devices into the optical field, combining high integration with high-speed processing and quantum computing [127–130]. Novel types of waveguides and optical fibers, new filters, high-speed switches, low-threshold micro-lasers, high-performance LEDs (light-emitting diode), photonic for VLSI (very large scale integration), along with smart materials applications like environmental, security, energy, transport, biological and chemical sensors are among the new opportunities in scientific and industrial areas like Information and Communication, Industrial Manufacturing and Quality, Life Sciences and Health, Emerging Lighting, Electronics and Displays, Security, Metrology and Sensors, Design and Manufacturing of Components and Systems, Agriculture and Food, Automotive and Transport [3].

Again in p. 13

Smart materials and Internet of Things (IoT)

Smart materials that can respond to external stimuli have been explored for a while now and by the virtue once this market is continuously growing new avenues and applications start to open for them to be used. More recently smart materials have been inserted on the concept – Internet of Things. This concept is in closely related with the project and idealization of Smart Cities where everything is connected providing to each individual the possibility of control, monitor and manage devices remotely [162]. Not only that, but also, IoT finds important application in industry, turning even more automotive the line of production, automotive, enabling the vehicles to be smarter and safer, healthcare, giving the possibility of accurate and quicker examinations and providing instant data to the doctors, retailing & logistics which can involve the use of less man power and consequently man interaction, security by turning even more precise and sensitive the security systems, and agriculture by developing more productive ways of treat and harvest the crops are some of the segments that IoT can be pretty successfully used.

Some typos in the text. Willey in page3, line 98, should be Wiley. Marked in page 3, line 140 may be market.

Thanks for this indication, the alteration has been made.

One of the main topics of this review article is why the inorganic glass is still promising. Therefore, the comparison of inorganic glass with other transparent materials, such as organic glass, transparent plastic, transparent papers, etc. should be carefully detailed.

The authors agree with the referee in that the comparison of inorganic glass with other transparent materials, such as organic glass, transparent plastic, transparent papers, etc. for opto-electronic and smart materials applications will enrich the manuscript while offer a holistic state of the art when transparent materials’ are concerned. In Closing Remarks some notes have been added concerning this issue.

Other possible substrates, such as plastic substrate and stretchable elastomers have been recently appearing as alternative to glass in the production of flexible displays and stretchable electronic devices and the performance of their displays compared taking into account parameters, such surface quality, transmission, and thermal and dimensional stability. These soft substrates are usually the natural rubber (NR), styrene butadiene rubber(SBR), ethylene-propylene-diene monomer (EPDM), polyurethane (PU), thermoplastic polyurethane (TPU) and predominant poly(dimethylsiloxane) (PDMS), etc., which can reversibly endure high deformations (>200%) [213]. However, the use of such materials often results in low electrical mobility and high electrical resistivity of electronic devices and mainly to a reduction in the working temperature.

Devices with good flexibility or stretchability based on a silicon membrane, single-walled carbon nanotubes, poly(ethylene naphthalate) film have been prepared and fabricated by standard methods on a carrier substrate like a Si wafer or a glass plate [214] or even directly on the flexible/elastic substrate, including low-temperature deposition, solution processing, nano-/micromolding, and electrospinning, etc. Currently being explored as an emerging technology is the use of transfer printing in the fabrication process of flexible and stretchable electronic devices. One example is the integration of small crystalline-silicon circuits (chiplets) in the active-matrix organic light-emitting-diode (OLED) displays [203,215]. Studies have shown that these chiplets could be transfer printed on the glass substrate via an elastomeric stamp to help build the integrated circuits. The OLED display is then formed and connected to the chiplets and demonstrating an exceptional performance and highlighting the effectiveness of the transfer printing method.

Reviewer 3 Report

This manuscript gives a detailed review about inorganic glass in smart materials and optoelectronic devices. It is well-written and be can accepted after minor revision.

1# In Table 5, the country of Corning should be USA. 

2# In Prospective Areas and Closing remarks, several sentences should be added to emphasize the possible alternative substrates, such as plastic substrate and elastomeric substrate for glass in flexible and stretchable electronic devices.

Author Response

Response letter

We did appreciate the Editorial Office and reviewers’ comments on the manuscript. We have revised the paper according to those suggestions and recommendations, which we do believe have improved the quality of the manuscript.

We believe we have addressed all the reviewer’s comments. The changes made have made in Track Changes mode, according to the Journal Guidelines.

We believe the manuscript has reached the Materials’ high publication standards, so we submit the manuscript to your final approval.

Sincerely yours,

MClara Gonçalves

#Reviewer 3

Comments and Suggestions for Authors

This manuscript gives a detailed review about inorganic glass in smart materials and optoelectronic devices. It is well-written and can accept after minor revision.

1# In Table 5, the country of Corning should be USA.

Thanks for this indication, the alteration has been made.

2# In Prospective Areas and Closing remarks, several sentences should be added to emphasize the possible alternative substrates, such as plastic substrate and elastomeric substrate for glass in flexible and stretchable electronic devices.

The authors agree with the referee in that the comparison of inorganic glass with other transparent materials, such as organic glass, transparent plastic, transparent papers, etc. for opto-electronic and smart materials applications will enrich the manuscript while offer a holistic state of the art when transparent materials’ are concerned. In Closing Remarks some notes have been added concerning this issue.

Other possible substrates, such as plastic substrate and stretchable elastomers have been recently appearing as alternative to glass in the production of flexible displays and stretchable electronic devices and the performance of their displays compared taking into account parameters, such surface quality, transmission, and thermal and dimensional stability.These soft substrates are usually the natural rubber (NR), styrene butadiene rubber(SBR), ethylene-propylene-diene monomer (EPDM), polyurethane (PU), thermoplastic polyurethane (TPU) and predominant poly(dimethylsiloxane) (PDMS), etc., which can reversibly endure high deformations (>200%) [213]. However, the use of such materials often results in low electrical mobility and high electrical resistivity of electronic devices and mainly to a reduction in the working temperature.

Devices with good flexibility or stretchability based on a silicon membrane, single-walled carbon nanotubes, poly(ethylene naphthalate) film have been prepared and fabricated by standard methods on a carrier substrate like a Si wafer or a glass plate [214] or even directly on the flexible/elastic substrate, including low-temperature deposition, solution processing, nano-/micromolding, and electrospinning, etc. Currently being explored as an emerging technology is the use of transfer printing in the fabrication process of flexible and stretchable electronic devices. One example is the integration of small crystalline-silicon circuits (chiplets) in the active-matrix organic light-emitting-diode (OLED) displays [203,215]. Studies have shown that these chiplets could be transfer printed on the glass substrate via an elastomeric stamp to help build the integrated circuits. The OLED display is then formed and connected to the chiplets and demonstrating an exceptional performance and highlighting the effectiveness of the transfer printing method.

Reviewer 4 Report

The manuscript is intended to review the subject. To my opinion the question in the title could be misunderstood. Inorganic glass could be used either as a supporting material or to be an active part in the development of a new application. For instance cover glasses are used in almost every photovoltaic module assembly. That means with growing sales of modules the demand for inorganic glass grows too. Without the need for innovations.

As I have learned from your examples however is that you analyze the potential of newly developed glasses with application specific attributes. Therefore I suggest to modify your title. May be like

“What is Driving the Development of New Inorganic Glass in Smart Materials and Opto-Electronic Devices?”

Throughout the manuscript please change “inorganic Glass” to “inorganic glass”.

Throughout the article I found the “advertisement” for “Corning Willow Glass” more than 15 times although in table 5 you list 19 more companies involved in the high-tech glass market

P3/line 141 ..academic star-ups.. please change to ..academic start-ups..

P3/line 144 Tables 4 e 5 change to Tables 4 and 5

Please reformat table 1

p2/268 ..near the absolute zero since… change to ..near zero temperature..

p6/380 ... imaginary, component χ"= χ sen …. imaginary, component χ"= χ sin phi

p9/509 ..absorbed h are also converted into electricity.. … absorbed h(ny) are also converted into electricity

p16/800 ... new applications of this this outstanding… ... new applications of this outstanding…

p17/818 ...will power groth and innovation… ...will power growth and innovation…

p17/848 ... History, 1st Editio, University of Chicago… ...History, 1st Edition, University of Chicago...

Author Response

Response letter

We did appreciate the Editorial Office and reviewers’ comments on the manuscript. We have revised the paper according to those suggestions and recommendations, which we do believe have improved the quality of the manuscript.

We believe we have addressed all the reviewer’s comments. The changes made have made in Track Changes mode, according to the Journal Guidelines.

We believe the manuscript has reached the Materials’ high publication standards, so we submit the manuscript to your final approval.

Sincerely yours,

MClara Gonçalves

#Reviewer 4

Comments and Suggestions for Authors

The manuscript is intended to review the subject. To my opinion the question in the title could be misunderstood. Inorganic glass could be used either as a supporting material or to be an active part in the development of a new application. For instance, cover glasses are used in almost every photovoltaic module assembly. That means with growing sales of modules the demand for inorganic glass grows too. Without the need for innovations.

As I have learned from your examples however is that you analyze the potential of newly developed glasses with application specific attributes. Therefore, I suggest to modify your title. May be like

“What is Driving the Development of New Inorganic Glass in Smart Materials and Opto-Electronic Devices?”

We would like to thank you for this suggestion but after a discussion we decided to maintain the original title once there are many articles in the text that are not based in new developments but in new applications of well-known synthesis routes and compositions.

Throughout the manuscript, please change “inorganic Glass” to “inorganic glass”.

Thanks for this indication, the alteration has been made.

Throughout the article I found the “advertisement” for “Corning Willow Glass” more than 15 times although in table 5 you list 19 more companies involved in the high-tech glass market.

That is true. Although a quite great number of companies are producing flexible glass, the fact is that Willow glass is by far the most used in academia and industrial applications when transparent flexible substrates are needed.

P3/line 141 ..academic star-ups.. please change to...academic start-ups…

Thanks for this indication, the alteration has been made.

P3/line 144 Tables 4 e 5 change to Tables 4 and 5

Thanks for this indication, the alteration has been made.

Please reformat table 1

We were not able to improve Table 1 design/format

p2/268 ..near the absolute zero since… change to ..near zero temperature.

Thanks for this indication, the alteration has been made.

p6/380 ... imaginary, component χ"= χ sen …. imaginary, component χ"= χ sin phi

Thanks for this indication, the alteration has been made.

p9/509 ..absorbed h are also converted into electricity.. … absorbed h(ny) are also converted into electricity

Thanks for this indication, the alteration has been made.

p16/800 ... new applications of this this outstanding… ... new applications of this outstanding…

Thanks for this indication, the alteration has been made.

p17/818 ...will power groth and innovation… ...will power growth and innovation…

Thanks for this indication, the alteration has been made.

p17/848 ... History, 1st Editio, University of Chicago… ...History, 1st Edition, University of Chicago...

Thanks for this indication, the alteration has been made.